# Influence of Strain and Diet on Urinary pH in Laboratory Mice

**DOI:** 10.3390/ani11030702

**Published:** 2021-03-05

**Authors:** Linda F. Böswald, Dana Matzek, Ellen Kienzle, Bastian Popper

**Affiliations:** 1Department of Animal Nutrition and Dietetics, Ludwig-Maximilians-Universität München, Schönleutnerstr. 8, 85764 Oberschleißheim, Germany; kienzle@tiph.vetmed.uni-muenchen.de; 2Biomedical Center, Core Facility Animal Models, Faculty of Medicine, Ludwig-Maximilians-Universität Munich, Großhaderner Straße 9, 82152 Planegg-Martinsried, Germany; dana.metzk@med.uni-muenchen.de

**Keywords:** metabolic acidosis, laboratory animals, mouse strains, DCAB

## Abstract

**Simple Summary:**

Dietary cation anion balance (DCAB) influences urine pH in several species, e.g., cats, dogs, pigs and cattle suggesting a species-specific impact. In the present study, we aimed to explore the impact of three diets with different DACB on wildtype laboratory mice. During a two-month feeding trial urine pH and body weight were measured in C57Bl/6J and CD1 male mice. Remarkable, we observed strong impact of the genetic background and diet on urine pH levels. A plausible explanation is that differences in renal phosphorus excretion and, in turn, phosphate buffering capacity account for these differences. It is tempting to speculate that standard laboratory mouse models show DCAB dependent variations in urine pH.

**Abstract:**

Acid base homeostasis and urine pH is influenced by the dietary cation anion balance (DCAB) in many species. Here, a negative DCAB acidifies the urine, while higher DCABs alkalize the urine. The dimension of the DCAB effect can be species-specific, because of differences in urine buffer systems. The aim of the present study was to describe the response of laboratory mice to diets with different DCAB. We used 8-week-old wildtype male mice of the C57Bl/6J inbred strain and CD1 outbred stock. Three groups (n = 15 animals/group) were formed and fed standard diet A for adaptation. For the 7-week feeding trial, mice were either kept on diet A (DCAB −7 mmol/kg dry matter (DM) or switched to diet B (246 mmol/kg DM) or C (−257 mmol/kg DM). Urine pH was measured weekly from a pooled sample per cage. There was a significant difference in the basal urine pH on diet A between C57Bl6/J and CD1 mice. The shift in urine pH was also significantly different between the two groups investigated.

## 1. Introduction

It has been shown in many species that the dietary cation anion balance (DCAB) influences acid base homeostasis and urine pH [1,2,3,4,5,6,7,8]. DCAB is calculated by adding the weighted amount of acidifying anions and alkalizing cations in the diet [5]. With the DCAB, the resulting urinary pH can be predicted with species-specific equations [1,3,4,5,6,7,8,9]. In cats and dogs, modification of urine pH is used in nutritional urolith prevention by altering the solubility of concrement-forming mineral salts [7,8,9,10]. A negative DCAB during late gestation is desired in dairy cattle to decrease the risk of hypocalcemia by sensitizing the endocrine control of calcium mobilization in an acidotic state [11]. A negative DCAB results in metabolic acidosis. Fed long-term, this can contribute to the reduction of bone mineral density [12,13] due to a PTH-mediated increase in renal calcium excretion [14,15]. Metabolic acidosis also induces renal phosphorus excretion, resulting in hypophosphatemia [16,17]. 

While the results of acidosis on bone metabolism and the kidneys have been investigated in laboratory mice [18,19,20] and mouse models for renal acid base homeostasis on a molecular level exist [21], comprehensive data about the effect of variations of DCAB on urine pH in laboratory mice is not yet available. In ruminants, individual differences in urinary phosphorus excretion have been observed [22,23]. These differences are attributed to genetic factors [22,24,25] determining the relationship between phosphorus absorption and plasma phosphorus concentration. In animals with higher intestinal phosphorus absorption, the renal excretion seems to become a more important pathway of excretion. Due to these aspects of genetic differences in renal mineral handling, we used an inbred and an outbred strain of laboratory mice to test for potential differences in response to the test diets. An acidifying effect of ammonium chloride has been reported in male outbred Crl:CD-1(ICR)BR mice [26]. It seems likely that mice react similar to shifts in DCAB as other species. In this case, a negative DCAB would lead to metabolic acidosis and respective metabolic consequences in those animals, e.g., in calcium and phosphorus homeostasis and bone metabolism. Consequently, several metabolic and urinary parameters will depend on DCAB. According to labelled information, the calculated DCAB of commercially available standard diets for laboratory rodents varies from negative to highly positive. Therefore, the objective of the present pilot study was to describe the effect of different DCABs on urine pH in laboratory mice. 

## 2. Materials and Methods

### 2.1. Animals

CD1 outbred mice (Crl:CD1(ICR), Charles river, Germany) and C57Bl/6J inbred animals (Charles river, Germany) were used in this study. Male CD1 and C57Bl/6J mice (8 weeks old) were allocated randomly to 3 diets per group (n = 15 animals/group) and housed in groups of 3 individuals per cage. 

### 2.2. Housing

The trial was approved of by the Ethical Committee of the Veterinary Faculty, Ludwig-Maximilians-Universität München (reference no. 178-06-19-2019). Housing of laboratory mice was in accordance with European and German animal welfare legislations (5.1-231 5682/LMU/BMC/CAM 2019-0007). Room temperature and relative humidity ranged from 20–22 °C to 45–55%. The light cycle was adjusted to 12 h light:12 h dark period. Room air was exchanged 11 times per hour and filtered with HEPA-systems. All mice were housed in individually ventilated cages (TypII long, Tecniplast, Germany) under specified-pathogen-free conditions. Hygiene monitoring was performed every three month based on the recommendations of the FELASA-14 working group. All animals had free access to water and food (irradiated, 10 mm pellet; 1314P, Altromin, Netherlands, in this study termed diet A). The cages were equipped with nesting material (5 × 5 cm, Nestlet, Datesand, UK), a red corner house (Tecniplast, Germany) and a rodent play tunnel (7.5 × 3.0 cm, Datesand, UK). Soiled bedding (LASbedding, 3–6 mm, PG3, Las vendi, Germany) was removed every 7 days. 

### 2.3. Feeding Trial

After one week of adaptation to the housing system and environment, during which the mice were fed the basal diet (diet A, DCAB −7 mmol/kg dry matter [DM]), the mice were either treated with diet B (V1154-000, Ssniff Spezialdiäten GmbH, Soest, Germany; DCAB 246 mmol/kg DM) or C (1814 Pmod., Altromin, Netherlands; DCAB −257 mmol/kg DM) or stayed on diet A for 7 additional weeks. Mice used in the present study were not killed at the end of the experiment. 

The diets were analyzed for minerals (Table 1) and the DCAB was calculated from these values using the following equation [1,3,4,6]: 

DCAB [mmol/kg DM] = 49.9 · Ca + 82.3 · Mg + 43.5 · Na + 25.6 · K − 59.0 · P − 62.4 · S − 28.2 · Cl; mineral content in g/kg DM.

### 2.4. Sampling

Once a week, during the cage changing, spontaneous urination is known to occur [27,28]. This behavior was used to collect as much urine as possible per mouse without stressful or invasive manipulation. Mice were taken from their old cage and if urine was voided during handling, it was directly collected in an Eppendorf tube (0.5 mL safe-lock tube, Ref.0030121023, Eppendorf, Germany). In detail, mice were restrained gently by grasping the loose skin on the neck. The fold was trapped between two fingers and the body was resting in the hand to support the animal’s weight. Mice were placed over the collection device and by light strokes of the animal’s belly, urine was released from the bladder. After that, mice were transferred into the cages. The urine was pooled per cage (3 animals) and pH was measured instantly using a glass electrode as part of the pH meter (PCE-PH 30, PCE, Germany; pH-Fix, Ref.92140, Macherey Nagel, Germany). Body weight (BW) was recorded once per week and the BW gain was calculated relative to the initial BW for each trial group. 

### 2.5. Statistics

Comparison between CD1 and C57Bl/6J mice was done via Student’s *t*-test in case of normal distribution and Mann–Whitney Rank Sum Test if normality testing (Shapiro–Wilk) failed. Comparisons between the three diet groups were conducted with a Kruskal–Wallis One Way Analysis of Variance on Ranks (SigmaPlot, Systat Software, San Jose, CA, USA). The significance level was set to *p* < 0.05. Data are presented as mean ± SD if not stated otherwise.

## 3. Results

There was a significant difference between initial and final BW between the strains (Table 2). BW gain in percent of initial BW differed between diets and strains. 

The basal urine pH of the C57Bl/6J inbred strain was significantly lower compared to CD1 outbred mice (5.7 ± 0.3 vs. 6.0 ± 0.2; *p* < 0.001; Figure 1). The urine pH values during the feeding trial were influenced by diet and strain (Table 3).

## 4. Discussion

In the present study, we used C57Bl/6J inbred and CD1 outbred mice to investigate whether there might be an effect of the genetic background on urinary pH in laboratory mice. For example, in ruminants, individual differences between animals of the same species are observed regarding urinary phosphorus excretion, possibly due to genetic differences [22,23,24,25]. Strains of laboratory mice show distinct physiological characteristics, so renal excretion of minerals like phosphorus might also be affected. Phosphate is an important buffer, thus differences in urine pH might be due to differences in phosphorus excretion. Indeed, we could show a significant difference between the urine pH of the C57Bl/6J inbred strain and CD1 outbred stock (mean ± SD 5.7 ± 0.3 vs. 6.0 ± 0.2; *p* < 0.001). while they were on the same basal diet (Figure 1). Mice responded in a similar way to the experimental diets in terms of urine pH shift (Table 3), but the C57Bl/6J mice had a significantly lower urine pH on diet B than the CD1 mice on diet B (6.3 ± 0.4 vs. 7.1 ± 0.3; *p* < 0.001).

There are numerous potential explanations for the differences in urine pH between C57Bl/6J and CD1 mice. There could be strain differences in the absorption of dietary cations and anions from the gastrointestinal tract. The activity of carboanhydrase, the enzyme for renal H^+^ excretion, plays a role in the potential change in urinary pH [4]. The possibility of strain differences in expression of this enzyme should be considered. Furthermore, buffering systems and the concentrations of acidifying or alkalizing compounds in the urine might differ between the strains. At this point, however, all explanations can only be speculative. Further investigations into the cause of the strain difference are necessary.

We used three commercially available diets for laboratory mice and found three rather different DCABs from negative to highly positive. Considering the effect of DCAB on urine pH found in this study, these diets are not comparable in terms of standardization. The metabolic effects differ between highly positive, nearly neutral, and negative DCABs so that e.g., urinary parameter or measurements on skeletal health cannot be compared between such diets. Labelling diets for laboratory animals with the DCAB value would clearly be valuable information for researchers planning experiments. Any parameter affected by acid base balance may not be comparable between the diets.

The present study could confirm that there is an influence of DCAB on urinary pH in laboratory mice, as known from other species. In general, a higher DCAB results in a higher urinary pH due to the higher concentration of alkalizing ions, as shown by diet B (Table 3). Diets with a negative DCAB lowered the urine pH in our study. From the data of the present study, it is not possible to know whether or not the resulting alkalosis or acidosis is compensated in C57Bl/6J and CD1 mice. In case there is a difference in their response to the DCAB, it can be speculated that among the multitude of laboratory mouse lines available, there will be a high variation in urine pH and its changes with DCAB.

BW development and steady gains according to the standard growth charts given by the animal breeder are important tools for researchers to monitor the health and welfare of their laboratory mice. It is known that genetic strains differ in their mean BW and their development, as the example of C57Bl/6J and CD1 mice in the present study also shows (Table 2). However, we observed an influence of diet with diet C mice having the highest BW gains, followed by diet A, then B, in both mouse strains (Table 2, *p* < 0.05). Acidifying diets can result in growth retardation [29]. In our study, the diet with the lowest BW gains had a positive DCAB, so that other reasons seem to be responsible for the differences in BW development. Differences in dietary energy content, palatability and feed intake as well as digestibility may be the reason for this. The percentage of starch and sugar in the carbohydrate fraction vs. other non-starch polysaccharides has an impact on digestibility [30]. The focus of this study was the effect of diet on urine pH, so we did not explore the aspect of BW development further. Nonetheless, the observation serves to demonstrate that diet influences BW development and deviations from the reference growth charts may be due to dietary factors.

Together, our results point towards an important role of diet for interpreting experiments. Only urine parameters measured from mice with the same genetic background should be compared. This has to be taken into account prior planning animal experiments, especially when comparing data from different studies. DCAB is an important parameter for numerous animal experiments in different fields including renal physiology, toxicology and pharmacokinetics. Therefore, widely used wildtype inbred and outbred strains have to be characterized regarding their diet behavior to increase reproducibility and interpretability of animal experiments.

## 5. Conclusions

The DCAB of three standard diets for laboratory mice differed markedly. There was a significant effect of diet on urine pH, with lower DCAB leading to lower urinary pH. Furthermore, we identified a significant effect of the genetic background on urinary pH. The cause of this, as well as possible consequences e.g., in bone turnover, need to be elucidated.

## Figures and Tables

**Figure 1 animals-11-00702-f001:**
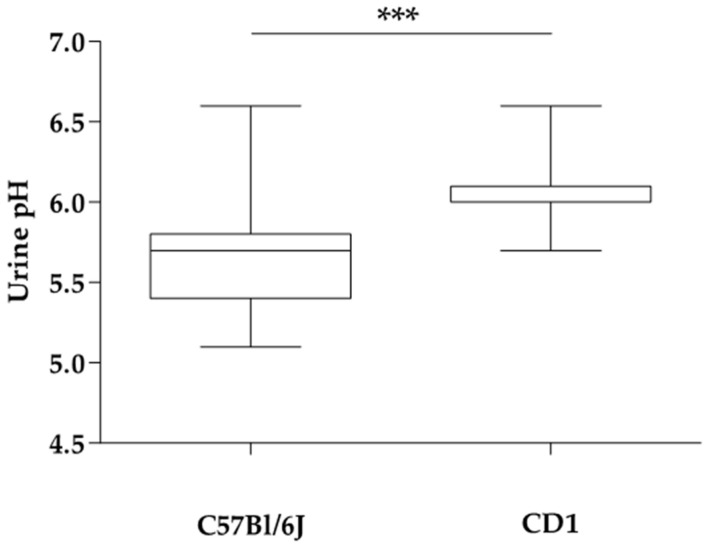
Basal urine pH of the C57Bl/6J inbred strain and the CD1 outbred stock (5.7 ± 0.3 vs. 6.0 ± 0.2; *** *p* < 0.001).

**Table 1 animals-11-00702-t001:** Analyzed nutrient content and calculated DCAB of the three test diets.

	Diet A	Diet B	Diet C
Dry matter (% as fed)	90.0	88.9	86.9
Calcium (% as fed)	0.62	0.96	0.61
Phosphorus (% as fed)	0.55	0.69	0.60
Magnesium (% as fed)	0.14	0.24	0.01
Potassium (% as fed)	0.56	0.80	0.39
Sodium (% as fed)	0.20	0.21	0.11
Chloride (% as fed)	0.53	0.50	0.46
Sulfur (% as fed)	030	0.33	0.32
DCAB (mmol/kg DM)	−7	246	−257

**Table 2 animals-11-00702-t002:** Body weight (BW) development of C57Bl/6J and CD1 mice on the three test diets.

	C57Bl/6J			CD1		
	Diet A	Diet B	Diet C	Diet A	Diet B	Diet C
Initial BW (g)	23.2 ^a^ ± 1.5	23.6 ^a^ ± 1.6	23.2 ^a^ ± 1.6	37.0 ^b^ ± 3.2	38.7 ^b^ ± 4.0	37.4 ^b^ ± 2.5
Final BW (g)	28.0 ^a^ ± 1.3	27.2 ^a^ ± 1.5	28.5 ^a^ ± 1.5	43.5 ^b^ ± 5.0	44.6 ^b^ ± 6.1	45.4 ^b^ ± 5.5
BW gain (% initial BW)	20.9 ^a,b^ ± 8.6	15.6 ^b,c^ ± 5.0	23.0 ^a^ ± 5.3	17.4 ^b,d^ ± 5.5	14.9 ^c^ ± 6.6	21.1 ^a,c,d^ ± 9.6

All data means ± SD. Different superscript letters within a line indicate significant differences (*p* < 0.05).

**Table 3 animals-11-00702-t003:** Urine pH values at the basal measurement and during the feeding trial.

	C57Bl/6J			CD1		
	Diet A	Diet B	Diet C	Diet A	Diet B	Diet C
Urine pH	5.7 ^a^ ± 0.2	6.3 ^b^ ± 0.4	5.6 ^b^ ± 0.2	6.3 ^c^ ± 0.5	7.1 ^a^ ± 0.3	5.6 ^b^ ± 0.2

All data means ± SD. Different superscript letters within a line indicate significant differences (*p* < 0.05).

## Data Availability

All relevant data is given in the paper. Additional information can be requested from the authors upon reasonable request.

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
