# Peer review of "Influence of Strain and Diet on Urinary pH in Laboratory Mice"

_animals, 2021, doi:10.3390/ani11030702_

Round 1
Reviewer 1 Report
Below are my comments: This is an interesting experimental study providing valuable information for planning of animal studies comprising dietary aspects. There are some minor points that that should either be added or commented. 1. Please describe in detail how the urine collection was carried out and the volume of collected samples. 2. It would be an advantage with some technical details of the pH-measurement. Please add information that this is a method based on glass electrode if that is so. 3. Inasmuch as the authors comment on the role of phosphate excretion; was it not possible to measure phosphate in the urine samples? Typographical/grammar: Page 2, line 56: “…similar to shifts in DCAB as other species” Page 6, line 167: “our” Page 6, line 167: “our”Author Response
Dear reviewer,
we appreciate the positive and constructive comments.
We are confident that we were able to address all points raised by you as a reviewer.
- Please describe in detail how the urine collection was carried out and the volume of collected samples.
Authors response: Thank you for pointing this out. We included a more detailed description on urine collection in the methods section. L. 97-105.
- It would be an advantage with some technical details of the pH-measurement. Please add information that this is a method based on glass electrode if that is so.
Authors response: We agree, based on the comment we added more detailed information about pH measurement in the methods section. L. 106-107.
- Inasmuch as the authors comment on the role of phosphate excretion; was it not possible to measure phosphate in the urine samples?
Authors response: Indeed, measuring urine P excretion would be the logical next step. We conducted a trial with mice of the same strains and measured P and creatinine concentration from pooled samples of spontaneous urine. Contrary to expectations, we did not find significant differences in P/crea excretion between the strains. Considering this new knowledge (measured after initial submission of the manuscript), we rewrote the section in the discussion, focusing more on other possible explanation for the urine pH differences. L.153-161.
The urine pH differences could be identically replicated in the P/crea trial, so they were not only coincidence.
In general, measuring minerals from mouse urine samples is challenging because of the small sample amounts spontaneously voided. Measuring P was our first aim, but proved not to be the cause for the observed differences. Information about other compounds in urine would be helpful, as now included in the discussion.
Typographical/grammar: Page 2, line 56: “…similar to shifts in DCAB as other species” Page 6, line 167: “our” Page 6, line 167: “our”
Authors response: Thank you for pointing this out. We included all recommendations in the revised version of the manuscript. l.57 and l.194 of the revised manuscript.
Reviewer 2 Report
Good work. No major concerns. Simple and straight to the point.
Author Response
Dear reviewer,
thank you very much. We appreciate the very positive feedback.
We hope that you will find our revised manuscript to be suitable for publication.
Reviewer 3 Report
Authors investigated influence of strain and diet on urinary pH. The results on urinary pH and body weight (BW) are interesting. I have some questions to develop authors’ discussion.
- Did authors measure urinary cation anion concentration? To conclude the difference of renal phosphorus excretion between C57Bl/6J and CD1, it needs urinary P concentration.
- Did authors measure urine volume? It may be a clue to discuss the difference of BW due to intake Na.
- Authors discussed about urinary pH due to urinary phosphate or carbonate. Do authors have any discussion about H+?
Author Response
Dear reviewer,
we appreciate the positive and constructive comments. As you will see from the rebuttal letter, we are confident that we were able to address all points raised.
- Did authors measure urinary cation anion concentration? To conclude the difference of renal phosphorus excretion between C57Bl/6J and CD1, it needs urinary P concentration.
Authors response: Thank you for pointing this out. The urinary CAB would be interesting to identify the source of the strain difference in urine pH. However, we did this investigation as a rather orientational study and used non-invasive methods for this. Collecting sufficient amounts of urine for CAB and/or daily total urine excretion would have meant housing the mice with restriction of movement for a longer period of time in metabolic cages (small urine amounts/day) to obtain enough urine.
In order to investigate the renal excretion of minerals or other potentially buffering substances, collection of larger amounts of urine would be valuable as a next step. The need to identify the compounds responsible for the pH differences is now discussed more in detail.
Did authors measure urine volume? It may be a clue to discuss the difference of BW due to intake Na.
Authors response: Urine volume collection would mean more invasive trials with housing in metabolic cages. This was not possible for the initial, orientational trial. The Na content in diets A and B was similar, diet C contained somewhat less Na, but not markedly so. All dietary Na contents met the requirements and did not pose a Na excess. Thus, we do not consider Na as major determinant for extremes in urine volume/concentration or pH.
- Authors discussed about urinary pH due to urinary phosphate or carbonate. Do authors have any discussion about H+?
Authors response: H+ may indeed be related to the urine buffering capacity. We included this in the discussion. L. 153-161.
Round 2
Reviewer 3 Report
I hope authors will investigate the mechanism of urine pH homeostasis.